# Validation of Challenges for the Development of the Marketing Plan for Startups Considering the Post-COVID-19 Reality: An Exploratory Analysis of the Brazilian Context Using Lawshe's Method

Ana Carla Magalhães Nascimento, Nathália de Kassia Galdino Oliveira, Verônica de Menezes Nascimento Nagata [ID], Reimison Moreira Fernandes and Vitor William Batista Martins *[ID]

Production Engineering Department, State University of Pará, 2626 Avenue Enéas Pinheiro, Belém 66087-670, Brazil; magalhaesanacarlanascimento@gmail.com (A.C.M.N.); nathaliadekassiaoliver@gmail.com (N.d.K.G.O.); vemenas@uepa.br (V.d.M.N.N.); reimison.fernandes@aluno.uepa.br (R.M.F.)
* Correspondence: vitor.martins@uepa.br

**Abstract:** Background: The post-COVID-19 scenario has demonstrated the increasing importance of marketing for organizations, as retailers and entrepreneurs have had to adapt to new ways of selling their products and services. In this regard, this research aimed to identify challenges for developing the marketing plan of startups and validate them from the perspective of managers in the field, considering the market characteristics inherent to the post-COVID-19 era; Methods: To achieve this, a literature review and a survey were conducted among professionals in the field. The collected data were analyzed using the quantitative Lawshe method. Results: The results indicate that, for the development of the marketing plan of startups considering the post-COVID-19 reality, it is important to prioritize overcoming the challenges of "Consumer behavior pattern change", "Differentiation from the competition", "Digital expansion", "Innovation capacity of companies", "Creation of transformative marketing", and "Reevaluation of marketing channels in the post-pandemic period"; Conclusions: Therefore, it can be concluded that these challenges reflect the main concerns and obstacles faced by startups in building effective marketing strategies and striving for a competitive position in the market. By recognizing and understanding these challenges, startups will be better prepared to face adversity and seize opportunities in this new market context.

**Keywords:** marketing plan; startups marketing plan; challenges in the marketing plan development; post-COVID-19 reality

## 1. Introduction

Currently, consumers have become actively engaged in co-creating their experiences with companies [1]. However, in the post-pandemic scenario, marketing becomes even more essential as retailers and entrepreneurs have had to adapt to selling their products/services online through the use of digital marketing. Additionally, consumer behavior is driven by the pursuit of goods and services that can help fulfill their higher-level social needs (e.g., social belonging and self-esteem) and self-actualization [2].

Marketing is based on a philosophy and ideology that demand efforts to identify and respond to the needs and desires of target markets better than competitors [2]. In this perspective, companies' adaptation to online sales with electronic attendants, applications, and methods to facilitate customer purchases is what ensures an advantage over others. As a result, all sectors have been impacted by this change; however, this change only accelerated an ongoing trend. As a result, industries that previously revolved around physical interaction have found ways and means to engage (and survive) through online platforms [2].

Regarding this, the decision-makers need to set realistic goals for their organizations during and after a crisis. In this regard, several companies have shifted their business models and invested capital in manufacturing personal protective equipment (as opposed to their regular products) to contribute to global efforts in combating COVID-19 [3–5] and to strengthen their branding by showcasing their contribution as a social agent to society. However, while many of the economic and health consequences of the pandemic are well documented, its marketing implications are less understood, as there is still limited literature specifically focused on this area.

Furthermore, Kotler and Keller [1] argue that through companies' marketing communication, they aim to inform, persuade, and remind the target consumer segment of the company's offerings. In this regard, marketing communication represents the "voice" of the company and its brands, serving as a means to establish dialogue and build relationships with and among consumers [1]. Regarding this, it is necessary for effective communication between the company and the target audience, especially in the post-pandemic context, to occur in both directions [1].

Concerning the context, this article aims to identify challenges for developing the marketing plan of startups through a review of the literature and validate them from the perspective of managers in the field, considering the market characteristics inherent to the post-COVID-19 era. In doing so, the article seeks to understand the problems faced during this period along with the strategies developed to strengthen the connection between marketing communication and customer attraction, satisfaction, and retention.

In addition to this introductory section, this article is structured into four other sections: First, a conceptual overview is presented, which discusses the definitions that establish the necessary theoretical foundation for the article. Following that, the methodological procedures are outlined, and research techniques and methods for the development of the study are presented. Subsequently, an analysis of the study's results is conducted, along with related discussions. Finally, the research conclusions are presented, followed by a list of references.

## 2. Background

In March 2020, the COVID-19 pandemic was declared a global concern by the World Health Organization (WHO), significantly impacting the daily operations of businesses and the global population [6,7]. Currently, in the post-pandemic period, as organizations adapt to contemporary technological methods to aid in the development of marketing plans and startups, there is a need for the creation of new integration channels and marketing strategies to ensure a competitive edge in the commercial sector. Consequently, numerous challenges emerge, hindering companies from easily achieving competitiveness and market penetration in the current landscape [8].

Regarding, studies have been conducted on the challenges that arise in the market due to the pandemic scenario, one of which is uncertainty in the market. Regarding this challenge, the authors emphasize that among the issues many marketing organizations face during pandemic lockdowns and the post-pandemic period are cash flow constraints, declining customer demand, disruptions in the supply chain, and managing remote operations, which accentuate market instability [8,9].

Some may argue that panic buying (including hoarding) is perfectly rational consumer behavior during crises like this with a significantly high level of uncertainty [2], given the ease and adaptability of customers to online shopping [10]. Therefore, organizations need to learn how to manage uncertainty to address such unpredictable and undesirable situations [11].

Furthermore, changes in customer behavior patterns also present a challenge. The COVID-19 crisis dramatically highlights these issues, as many small businesses worldwide have simply collapsed due to a lack of demand from their regular customers and the inability to adopt alternative ways of doing business [11]. Thus, what was once focused on the product has shifted towards being customer-centric, focusing on their needs. As a

result, companies have had to adapt and find ways to manage their inventory to remain competitive in the market [12]. Therefore, the most significant change for an existing organization aiming to become customer-centric is the addition of one or more solution units [12]. As a result, the pursuit of meeting customer needs and the behavioral shift, especially in the pandemic scenario, have necessitated companies to modify themselves to address the new demand and cope with market instability [3].

Some researchers have viewed competition as a Yin and Yang concept, where if opposing cooperative and competitive forces are balanced, they can be advantageous [3]. Moreover, the search for products that have a lesser environmental impact and demonstrate care for the environment has been highly demanded and serves as a form of differentiation. This is because environmentally conscious or green consumers have become numerous and represent a valuable segment for many companies [13]. In addition, the pressure from communities, the media, NGOs, environmental groups, and labor unions, as well as consumer behavior, can also drive organizations towards sustainable practices [14].

However, intensifying competition requires companies to differentiate themselves even more. This means that investment in technology and marketing is of paramount importance in this process. Over the past two decades, social media technologies have evolved in their purpose of use, from facilitating individual expression through online word-of-mouth (WOM) to their more recent role as a source of market intelligence for building brand experience through increased customer engagement [15]. With that in mind, considering that technologies are advancing and consumers read at least four reviews before making a purchase decision, it becomes crucial for businesses to adapt to this trend [16]. It is important to ensure quality and commitment to the customer because the use of social media also generates a significant increase in customer insights, including how consumers are interacting with each other and the products and services they consume [16]. Thus, being present in the digital realm and embracing new technologies and advancements is of utmost importance.

This process also requires proper strategic preparation, establishing trust, process-oriented thinking, integration and reinforcement of all involved parties, separate yet collaborative knowledge, and organizational alignment [17]. As it provides new and valuable sources of information that describe how customers gather information, use that information, and how it influences their decision-making, purchasing behavior, and post-purchase behavior, embracing digital platforms creates opportunities for improvement for companies across all sectors [16]. This enables the analysis of the ability to generate and leverage deep customer insights, manage brand health and reputation in a marketing environment where social media plays a significant role, and evaluate the effectiveness of digital marketing efforts. [16]. However, while the transformation process was smooth for some institutions, others responded with a crisis-driven migration process due to the pandemic [17]. As a result, those who were not connected to the internet and lacked digital skills faced complete exclusion in this scenario [18]. As a consequence, the closure of numerous micro and small businesses occurred during the pandemic due to their inability to adapt to the digital landscape.

However, changes in sanitary regulations also bring about another set of challenges, as the COVID-19 pandemic had a devastating impact on global society and the world economy, fundamentally altering the international economy and all human activities [19]. Therefore, companies are making various adjustments to ensure a balanced approach to data privacy and health protection in the context of COVID-19 [20]. As a result, the main challenge was adapting to new work routines to avoid mass layoffs or business closures. To address this, several measures were adopted, with one prominent example being remote work or a home office. In this setup, companies provide the necessary resources to their IT departments to support employees working securely from home. Dedicated support and training in risk and mitigation measures for remote work are also implemented, including clear and continuous communication. Additionally, organizations include a data privacy leader in their COVID-19 response team to ensure early assessment and discussion of potential

measures that may affect data privacy [20]. All of these measures are implemented as a way to introduce solutions in industrial enterprises to minimize the possibility of COVID-19 virus transmission and ensure the safety of employees on the premises [19].

The challenge of innovation capacity in companies is also noteworthy as a relevant factor for their recovery in the post-pandemic period. In this regard, companies were compelled to quickly innovate their products to adapt to changing market demands and consumer preferences [21] and sustain their businesses [22]. Thus, innovation through the development or adaptation of new products enabled companies to increase their sales volume by catering to different customer segments, thereby resolving their cash flow issues [23]. Furthermore, the increased market uncertainty has also heightened the need for the short-term adoption of advanced technologies [24]. Improving the innovation performance of companies, these technology investments have created opportunities for businesses to quickly adapt to the changes occurring during the COVID-19 pandemic and redesign the ways they deliver their offerings [25]. However, the different innovation strategies implemented by companies also had varying impacts and reactions, both positive and negative. This is because the market has varying levels of maturity that require different levels of readiness from companies [26].

Moreover, the difficulty of adaptation by some consumers to online shopping could also influence the limited reach of companies for these individuals, as the internet has become one of the primary sources of information. Consequently, many studies on offline and online shopping have primarily focused on key consumer components such as demographics, motivations, and buying patterns. They primarily examine how consumer buying patterns can be influenced by demographics and motivations [27]. Thus, individuals with lower levels of education, older adults, and those with limited access to technology and the Internet tend to face more challenges in making online purchases due to their physical or social limitations [28]. Despite these challenges, the online shopping sector has steadily grown over the past decade, particularly in the post-pandemic period, and has become an important area of study in consumer research [29].

The creation of transformative marketing is also a challenge faced by companies, as the market constantly evolves, demanding that businesses continually innovate their strategies. As a result, transformations in the marketing function reflect changes in the immediate business environment, and their impacts can be identified through the financial results of companies. In other words, the constant changes among consumers, markets, and marketing departments play a significant role in the need for business transformations [30]. In this sense, transformative marketing recognizes the importance of agile and informed actions in response to a crisis, as untimely and uninformed actions can result in greater losses and missed opportunities [9]. Additionally, transformative marketing also promotes a value proposition that does not harm others and should integrate transformative change that creates value for individuals and society, as well as for the natural environment [31].

Another important point is the challenge of creating strategies to combat the economic damage caused by the pandemic. In the context of today's interconnected world, the economic impact of COVID-19 on different sectors is interrelated, and the problems are globally interconnected. Thus, the lockdowns and restrictions implemented during this period in different parts of the world significantly reduced the production of goods and services. As a result, there was a global slowdown in economic activity, along with restrictions on the movement of products that adversely affected the entire supply chain, resulting in market disruptions [32]. In this way, when looking at the financial market, it is observed that COVID-19 has increased the risk of global financial collapse, negatively impacting the global financial market [33]. Thus, for businesses, there has been a definitive and significant impact on capital costs to survive and attract potential investments. Governments worldwide, which also exert significant pressure on the global economy, have been unable to fully mitigate and protect the financial system, resulting in a significant limitation of financial resources and leading to a global crisis [34].

Additionally, the challenge of reassessing marketing channels is also noteworthy, as they had to be reconsidered to adapt to market changes, especially in light of the pandemic crisis. Since the widespread adoption of mobile devices, consumers have gained access to information and can choose what and where to purchase, whether it be through digital or physical retail platforms, in real time. This has revolutionized the concept of multichannel purchasing [35]. As a result, omnichannel customers can utilize all the omnichannel resources of companies but on different occasions or buying journeys, meaning they are single-channel at any specific purchasing occasion [36]. However, this consumer behavior has forced all multichannel companies to consider how to manage and influence interactions with customers across various touchpoints and channels to leverage their business [37].

Building customer trust is indeed a significant challenge for businesses, especially in the post-COVID-19 era, considering the points mentioned earlier. In this context, the fear of contagion and changing financial conditions have altered consumer buying behavior. As a result, there has been an even greater market demand for transparency in business practices, higher product or service quality, more personalized customer service, and the reputation and credibility of companies. This has led consumers to consider purchasing methods that further reduce risks [38]. In this scenario, it has been observed that when consumers have greater perceived control over their interactions with companies, it can positively affect the trust they place in those companies [39]. However, it is worth noting that the sharing of information and the act of making purchases by consumers are directly related to the perception of benefits obtained in times of need. Therefore, it has become essential for companies to demonstrate credibility to customers to gain their trust and foster customer loyalty [40]. Table 1 presents a summary of the main challenges identified in the literature and considered in this study.

**Table 1.** Challenges in the marketing plan development.

| Code | Challenges | References |
|---|---|---|
| D1 | Uncertainty in the face of the market | [2,8–11,41] |
| D2 | Changing pattern of consumer behavior | [3,11,12] |
| D3 | Need to differentiate from the competition | [3,13,14] |
| D4 | Increased competitiveness | [15,16] |
| D5 | Changes in health regulations and standards | [19,20] |
| D6 | Expansion in the digital environment | [16–18] |
| D7 | Innovation capacity of companies | [21–26] |
| D8 | Difficulty for consumers to adapt to online shopping | [27–29] |
| D9 | Creating transformative marketing | [9,30,31] |
| D10 | Creation of strategies to combat the damage caused after the pandemic | [32–34] |
| D11 | Reassessment of marketing channels | [35–37] |
| D12 | Building customer trust | [38–40] |

Source: Authors based on the literature presented in the table.

## 3. Methodological Procedures

To conduct the present research, four stages were developed, namely: (a) a literature review on the challenges for developing the marketing plan of startups considering the post-COVID-19 reality; (b) the development of a questionnaire based on the challenges identified in the literature; (c) the administration of the questionnaire to marketing professionals (survey), as well as the use of the Lawshe method to analyze the validity of these challenges in the analyzed context; and (d) the elaboration of discussions conducting a critical analysis in light of the literature in the field, highlighting theoretical and practical implications, as well as drawing conclusions on the achieved results.

Firstly, it is worth highlighting the scientific databases considered during the literature review phase: Science Direct, Web of Science, Scopus, Springer, and Wiley. This phase consisted of acquiring prior knowledge on the topic of challenges for developing the marketing plan of startups considering the post-COVID-19 reality. In the searches conducted in the aforementioned databases, specific terms and combinations were used, such as: marketing, marketing and pandemic, marketing strategy, marketing communication, marketing solution, marketing challenges, marketing plan, marketing barriers, channel marketing, and consumer loyalty marketing. As a result, 12 challenges were identified (as shown in Table 1).

Subsequently, a questionnaire was developed based on the identified challenges, and experts were asked to evaluate each one using three response options: 1—The consideration of this challenge is essential for developing the marketing plan of startups considering the post-pandemic scenario; 2—The consideration of this challenge is important but not essential for developing the marketing plan of startups considering the post-pandemic scenario; and 3—The consideration of this challenge is not important for developing the marketing plan of startups considering the post-pandemic scenario.

A total of 85 invitations were sent to marketing professionals through social media platforms such as LinkedIn, WhatsApp, and email. The survey was conducted using Google Forms, which generated a link for respondents to complete the questionnaire. The response rate was 34.11%. It is important to note that all participating professionals work in the field of Marketing in the Brazilian market. Among the respondents, 79% were from the North, 3% from the Northeast, 10% from the South, and 7% from the Southeast region. Regarding their positions, 14% were analysts, 17% were assistants and associates, 41% were CEOs, coordinators, and directors, 10% were interns, filmmakers, and trade marketers, and 17% were managers. In terms of experience, 62% had less than 5 years in the market, 31% had between 5 and 10 years of experience, and 10% had over 10 years of experience in the market.

Afterward, the collected data was processed using the Lawshe method. This method was employed to analyze and validate the challenges identified in the research. This phase was carried out following the guidelines presented by [42,43]. At first, the Content Validity Ratio (CVR) was calculated for each criterion in the questionnaire. The CVR values range from −1 to +1, where −1 indicates total disagreement and +1 indicates total agreement. When evaluating the unfavorable results obtained from the procedure, it is worth noting that a positive CVR was obtained when more than 50% of the respondents considered the analyzed items "essential". Conversely, a negative CVR is obtained when less than 50% of the respondents deem the analytical item "non-essential". When the CVR is equal to 0, it indicates that half of the experts consider the criterion "necessary" while the other half does not [44]. Then calculations are performed. So, the calculation of $CVR_{critical}$, is applied to analyze items that can be discarded in the final composition, due to CVR values below the critical limit. For the calculation in $CVR_{critical}$, the parameters mean, variance, and standard deviation are considered. All the aforementioned equations are presented in detail in Figure 1.

After the calculations were performed, debates and discussions were conducted on the obtained results, analyzing them critically in light of the literature in the field. Subsequently, the contributions to theory and practice were highlighted, along with the conclusions and directions for future research.

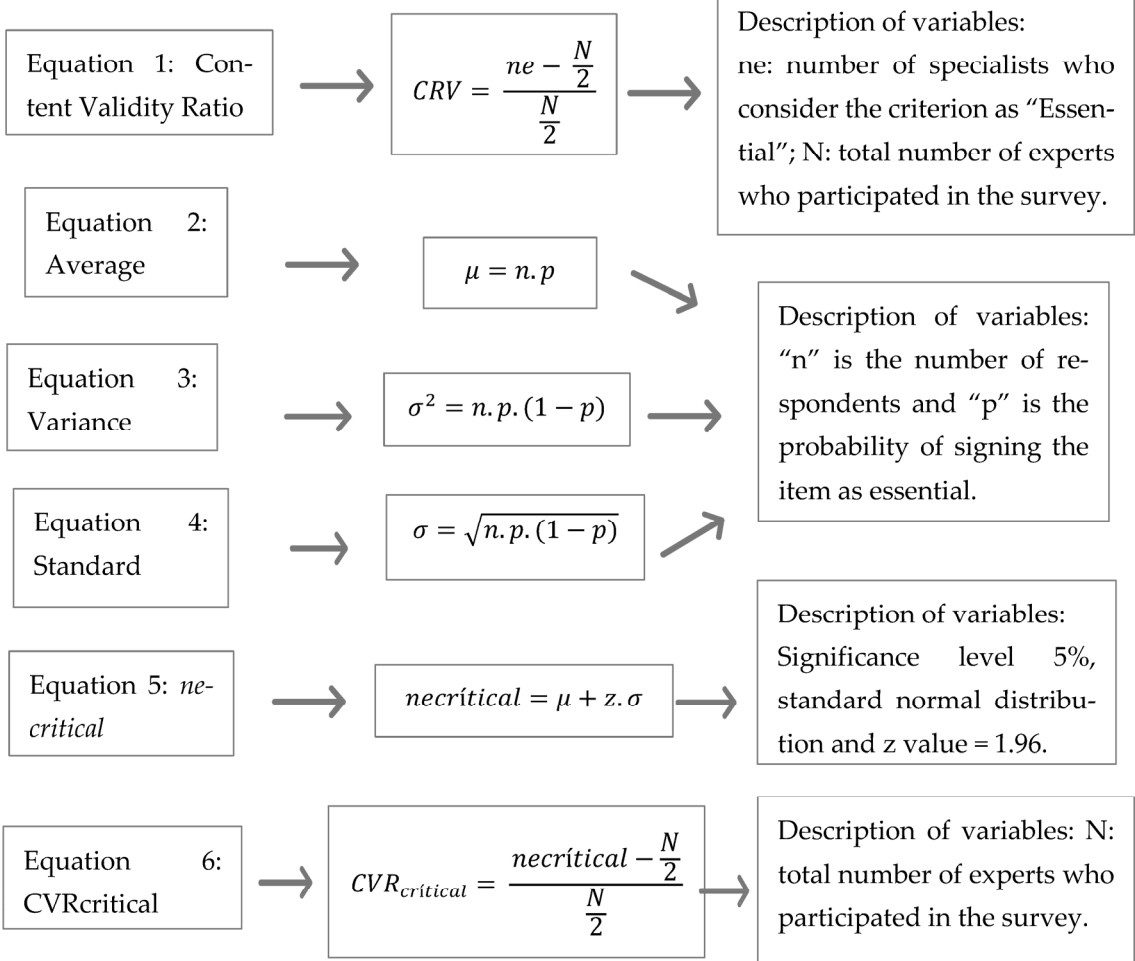

**Figure 1.** Lawshe method equations.

## 4. Results

Based on the methodological procedures described in the previous section, the Response Validity Coefficient (CVR) was calculated for each challenge addressed in the research, and then the $CVR_{critical}$. It is important to point out that the total sample in the calculations of this study consisted of 29 marketing specialists working in Brazil. The $CVR_{critical}$ obtained was 0.364, and this value was used as a reference in the validation analysis of the challenges. The challenges that presented a CVR coefficient above 0.364 were considered valid in the study context. On the other hand, the challenges that obtained a lower CVR coefficient were considered invalid. Table 2 presents the results achieved in this study.

**Table 2.** Validation of challenges via Lawshe's method.

| Code | Number of "Essential" Responses | Content Validity Ratio (CVR) | $CVR_{critical}$ Validation Reference: 0.364 |
|---|---|---|---|
| D1 | 18 | 0.241 | X |
| D2 | 26 | 0.793 | ✔ |
| D3 | 23 | 0.586 | ✔ |
| D4 | 15 | 0.034 | X |
| D5 | 14 | −0.034 | X |
| D6 | 27 | 0.862 | ✔ |

**Table 2.** *Cont.*

| Code | Number of "Essential" Responses | Content Validity Ratio (CVR) | $CVR_{critical}$ Validation Reference: 0.364 |
|---|---|---|---|
| D7 | 22 | 0.517 | ✔ |
| D8 | 17 | 0.172 | X |
| D9 | 20 | 0.379 | ✔ |
| D10 | 18 | 0.241 | X |
| D11 | 23 | 0.586 | ✔ |
| D12 | 18 | 0.241 | X |

Source: Authors (2023).

*4.1. Associated Debates*

Observing the data presented in Table 2, it is possible to identify the challenges that were validated based on the opinions of specialists in the marketing sector who work in the Brazilian context. Therefore, for the elaboration of the marketing plan for startups considering the post-pandemic reality of COVID-19, it is important to prioritize overcoming the challenges of "Change in the pattern of consumer behavior", "need to differentiate about the competition", "expansion in the digital environment", "innovation capacity of companies", "creation of transformative marketing" and "reassessment of marketing channels in the post-pandemic".

Changing patterns of consumer behavior Regarding the challenge of changing consumer behavior patterns, it is relevant to highlight its overcoming, as the COVID-19 pandemic has changed the entire landscape from what it used to be. Even with the capacity and necessity to understand and manage business uncertainties, there is no established framework to deal with all uncertainties, as emphasized by [11] and reinforced by [3]. Marketing can be affected in various ways, as it generates increased competition among companies, resulting in greater investment in differentiation. However, marketing strategies, being in constant evolution, force companies to adopt innovative strategies to stay competitive in the market. One such strategy is the addition of customer-centric solutions, as mentioned by [12], since after the pandemic, customer support services and communication channels needed to be more incisive.

Another important consideration is the need to differentiate organizations from the competition, since many companies have closed their doors and bankruptcy rates have increased due to the impact suffered by the COVID-19 pandemic. With that, there is a need for differentiation to ensure competitiveness and remain inserted in the market, as mentioned in [3]. In this way, marketing innovation helps companies obtain competitive advantages, according to [14]. Additionally, the expansion of organizations into the digital environment was important to ensure visibility during the quarantine period and, as previously mentioned, to ensure competitiveness in the market environment. However, with the migration to the digital environment, challenges emerged, such as Internet instability, a lack of technological resources, and limited knowledge in informatics, as stated by [17].

The capacity for innovation also behaves as a relevant challenge to be overcome by organizations. As emphasized by [21], companies were driven to rapidly innovate their products as a strategy to sustain their businesses during the pandemic. Innovation, whether through the development of new products or the adaptation of existing ones, played an important role in increasing sales volume and serving different customer segments, allowing them to solve their financial problems. In addition, market uncertainty has also increased the need to adopt advanced technologies in the short term, improving companies' innovation performance, as highlighted by [24]. These technology investments have created opportunities for companies to quickly adapt to changes during the COVID-19 pandemic and redesign how they deliver their offerings.

In addition, the creation of transformative marketing emerges as an important challenge for companies, since the market is constantly sophisticated, demanding that organizations constantly innovate in their strategies. As pointed out by [30], changes in consumers,

markets, and marketing departments play a significant role in the need for business transformations, as reflected in identifiable impacts on companies' financial results. In this sense, transformative marketing recognizes the importance of agile and informed actions in response to crises, avoiding untimely and uninformed actions that can lead to significant losses and opportunity costs [9]. In addition, transformative marketing seeks to promote a value approach that does not harm other stakeholders, integrating transformative change that creates value for individuals, society, and the environment.

Another important point is the reassessment of marketing channels, especially in the post-pandemic context. As highlighted by [35], consumers now have access to real-time information and can choose where and how to shop, revolutionizing the multi-channel shopping experience. In this regard, companies need to develop effective strategies to manage and influence interactions with consumers across multiple touchpoints to drive their business forward.

### 4.2. Implications for Theory and Practice

The achieved results have implications for both theory and practice. From a theoretical perspective, the findings of this study have important implications for the field of marketing, especially in the post-pandemic context. By exploring the challenges faced by startups in developing their marketing plans, this study contributes to the advancement of the literature and aligns with previous research [21]. The study's focus on innovation and validation of these challenges provides valuable insights into the specific demands that these companies face, greatly contributing to their recovery. The validated challenges also align with existing literature by emphasizing the importance of overcoming these aspects for the success of startups in the current landscape. Furthermore, the study offers important approaches to address the necessary changes in marketing strategies to navigate ongoing socioeconomic and technological transformations. These findings enrich the theoretical understanding of marketing practices in the current context and establish a solid foundation for overcoming the challenges faced in constructing startup marketing plans.

From a practical standpoint, the study conducted helps organizations gain a better understanding of how to deal with challenges and uncertainties during scenarios like the COVID-19 pandemic. This information assists companies in ensuring their survival and effectively harnessing the power of innovation and differentiation, thus achieving competitiveness and market presence. Furthermore, it provides a foundation for marketing managers to develop plans that align with the new challenges present in the market, taking into account the specificities of startups.

### 5. Conclusions

Based on the achieved results, it was possible to identify and validate challenges in the development of marketing plans for startups, considering the post-COVID-19 market context. These findings provide a significant contribution to the understanding of the difficulties and obstacles faced by startups in this new reality, offering insights that can assist them in developing effective marketing strategies and pursuing differentiation and competitiveness in a landscape characterized by constant transformations and uncertainties.

This study has identified and validated significant challenges for startups in the development of their marketing plans in the post-pandemic context. These challenges have proven crucial for the success and survival of these companies as they strive to adapt to accelerated changes and new market demands. Among the analyzed and validated challenges in the research, noteworthy ones include the change in consumer behavior patterns, the need for differentiation from competitors, expansion in the digital realm, the capacity for innovation within the companies, the establishment of transformative marketing practices, and the reevaluation of marketing channels in the post-pandemic era.

The challenges identified reflect the main concerns and obstacles faced by startups in building efficient marketing strategies and striving for a competitive position in the market.

By recognizing and understanding these challenges, startups will be better prepared to face adversity and capitalize on the opportunities that arise in this new context.

It is important to note that this study has some limitations. The main limitation is related to the exploratory nature of the research, which implies a specific and restricted sample of marketing specialists operating in Brazil. Therefore, although a diverse sample of professionals from the field was obtained, including specialists from different regions with varied positions and experiences, it is important to consider that the results cannot be generalized to other contexts with different specificities.

However, despite the limitations mentioned, this research provides a solid foundation for future investigations in this area. Based on the results obtained and the identified gaps, several proposals for future research can be made. Firstly, it would be interesting to conduct a longitudinal study to track the evolution of the identified challenges over time, allowing for a deeper understanding of changes and trends in the post-pandemic context. Additionally, it would be valuable to investigate the specific strategies and approaches adopted by successful startups to overcome these challenges, aiming to identify best practices and insights that can be shared with other companies in similar situations. Another relevant point would be to explore the role of partnerships and collaborations between startups, established companies, and support institutions in addressing challenges and fostering innovation in the marketing sector. Lastly, considering the growing importance of corporate social responsibility, future research could explore the influence of sustainability and social responsibility practices on the marketing strategies of startups, particularly in the post-pandemic context. Such future research can contribute to the improvement of marketing strategies for startups and the development of an environment more conducive to their growth and success.

In conclusion, this study provides an important contribution to understanding the challenges faced by startups in developing their marketing plans in the post-pandemic context of COVID-19. However, there is room for further research to deepen and expand knowledge on the topic. Some suggested future studies include: (a) the development of an action plan for defining marketing strategies for startups based on the validated challenges in this study; (b) the implementation of marketing strategies through action research in selected startups; and (c) the proposal of a set of marketing indicators specific to startups in the post-pandemic period of COVID-19. These future studies can provide valuable insights and practical guidance for startups navigating the evolving marketing landscape.

**Author Contributions:** Conceptualization, A.C.M.N. and N.d.K.G.O.; methodology, V.W.B.M.; software, V.W.B.M.; validation, V.W.B.M., R.M.F. and V.d.M.N.N.; formal analysis, V.W.B.M.; investigation, N.d.K.G.O.; resources, A.C.M.N.; data curation, N.d.K.G.O.; writing—original draft preparation, N.d.K.G.O. and A.C.M.N.; writing—review and editing, V.W.B.M.; visualization, V.d.M.N.N.; supervision, R.M.F.; project administration, V.W.B.M. All authors have read and agreed to the published version of the manuscript.

**Funding:** This research received no external funding.

**Institutional Review Board Statement:** The study did not require one.

**Informed Consent Statement:** Not applicable.

**Data Availability Statement:** https://drive.google.com/file/d/1Fd7ViKKUFAENsR2Zkb5kk8GZEeXt5Kbz/view?usp=drive_link (accessed on 4 July 2023).

**Conflicts of Interest:** The authors declare no conflict of interest.

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
