# Peer review of "Validation of Challenges for the Development of the Marketing Plan for Startups Considering the Post-COVID-19 Reality: An Exploratory Analysis of the Brazilian Context Using Lawshe’s Method"

_fintech, doi:10.3390/fintech2030032_

Round 1

Reviewer 1 Report

The manuscript presented is a topic of current interest, uses contemporary references, and is transparent in providing its data. Still, improvements are needed.

1) The introduction needs to be adjusted, to give a more logical line to the text and to lead it more directly to the aim.

2) The background digresses too much, at times it moves away from the central idea, it is better to summarize what is relevant.

3) Lines 78 to 84?

4) Table 1 is a result?. Why is it not part of the results section?

5) It is important to know how the articles are selected to generate the 12 challenges (D1 ... D12). All the following depends on this. Is it the product of a systematic review? Is it the product of a bibliometric analysis? If not at least explain and state limitations of possible selection bias.

6) A discussion section is needed, highlighting the contribution of the manuscript in a separate section from the results.

Author Response

Rebuttal Letter

Manuscript ID: fintech-2533358

Dear reviewer,

Thank you very much for the considerations presented. These considerations have enabled a considerable improvement in our manuscript. We adjust according to your considerations. We list below all the considerations made and justify the actions taken. Thanks for your precious help.

Rebuttal items

Reviewer #1: The manuscript presented is a topic of current interest, uses contemporary references, and is transparent in providing its data. Still, improvements are needed:

Reply: Thanks for the comment.

1) The introduction needs to be adjusted, to give a more logical line to the text and to lead it more directly to the aim.

Reply: Thanks for the comment. We have excluded some points from the introduction in order to make the information more direct to the main objective. In addition, we tried to make a better transition between paragraphs. Please check the new version of the article.

2) The background digresses too much, at times it moves away from the central idea, it is better to summarize what is relevant.

Reply: Thanks for the comment. Parts of the text that are more distant from the central idea were excluded. Please check the new version of the article.

3) Lines 78 to 84?

Reply: Thank for the comment. It was excluded. This part is a residue of text left over from the journal formatting template. We apologize for what happened. Was removed. Please check this new version of the paper.

4) Table 1 is a result? Why is it not part of the results section?

Reply: Thank for the comment. Table 1 is the result of the literature review carried out. It presents a summary of the challenges in preparing the startups marketing plan presented in the background section 2. Therefore, it was found more convenient to present it at the end of section 2 as a section closure.

5) It is important to know how the articles are selected to generate the 12 challenges (D1 ... D12). All the following depends on this. Is it the product of a systematic review? Is it the product of a bibliometric analysis? If not at least explain and state limitations of possible selection bias.

Reply: Thank for the comment. How the articles were selected is highlighted in the methodological procedures section (see below).

“Firstly, it is worth highlighting the scientific databases considered during the literature review phase: Science Direct, Web of Science, Scopus, Springer, and Wiley. This phase consisted of acquiring prior knowledge on the topic of challenges for developing the marketing plan of startups considering the post-COVID-19 reality. In the searches conducted in the aforementioned databases, specific terms and combinations were used, such as: marketing, marketing and pandemic, marketing strategy, marketing communication, marketing solution, marketing challenges, marketing plan, marketing barriers, channel marketing, and consumer loyalty marketing. As a result, 12 challenges were identified (as shown in Table 1).”

6) A discussion section is needed, highlighting the contribution of the manuscript in a separate section from the results.

Reply: Thank you for the comment. There is a discussion section within section 4, as indicated by the journal (see below).

4.1. Associated debates

Observing the data presented in Table 2, it is possible to identify the challenges that were validated based on the opinion of specialists in the marketing sector who work in the Brazilian context. Therefore, for the elaboration of the marketing plan for startups considering the post-pandemic reality of COVID-19, it is important to prioritize overcoming the challenges of “Change in the pattern of consumer behavior”, “need to differentiate about the competition”, “expansion in the digital environment”, “innovation capacity of companies”, “creation of transformative marketing” and “reassessment of marketing channels in the post-pandemic”.

Regarding the challenge of changing consumer behavior patterns, it is relevant to highlight its overcoming, as the COVID-19 pandemic has changed the entire landscape from what it used to be. Even with the capacity and necessity to understand and manage business uncertainties, there is no established framework to deal with all uncertainties, as emphasized by [11] and reinforced by [3]. Marketing can be affected in various ways, as it generates increased competition among companies, resulting in greater investment in differentiation. However, marketing strategies, being in constant evolution, force companies to adopt innovative strategies to stay competitive in the market. One such strategy is the addition of customer-centric solutions, as mentioned by [12], since after the pandemic customer support services and communication channels needed to be more incisive.

Another important consideration is regarding the need to differentiate organizations from the competition, since many companies have closed their doors and bankruptcy rates have increased due to the impact suffered by the COVID-19 pandemic, and with that, there is a need for differentiation to ensure competitiveness and remain inserted in the market as mentioned by [3]. In this way, marketing innovation makes companies obtain competitive advantages according to [15]. Additionally, the expansion of organizations to the digital environment was important to ensure visibility during the quarantine period and, as previously mentioned, to ensure competitiveness in the market environment. However, with the migration to the digital environment, challenges emerged such as: Internet instability, few technological resources and limited knowledge in informatics, as stated by [18].

The capacity for innovation also behaves as a relevant challenge to be overcome by organizations. As emphasized by [22], companies were driven to rapidly innovate their products as a strategy to sustain their businesses during the pandemic. Innovation, whether through the development of new products or the adaptation of existing ones, played an important role in increasing sales volume and serving different customer seg-ments, allowing them to solve their financial problems. In addition, market uncertainty has also increased the need to adopt advanced technologies in the short term, improving companies' innovation performance, as highlighted by [25]. These technology investments have created opportunities for companies to quickly adapt to changes during the COVID-19 pandemic and redesign how they deliver their offerings.

In addition, the creation of transformative marketing emerges as an important challenge for companies, since the market is in constant sophistication, demanding that organizations constantly innovate in their strategies. As pointed out by [31], changes in consumers, markets and marketing departments play a significant role in the need for business transformations, reflected in identifiable impacts on companies' financial results. In this sense, transformative marketing recognizes the importance of agile and in-formed actions in response to crises, avoiding untimely and uninformed actions that can lead to significant losses and opportunity costs [9]. In addition, transformative marketing seeks to promote a value approach that does not harm other stakeholders, integrating transformative change that creates value for individuals, society, and the environment.

Another important point is the reassessment of marketing channels, especially in the post-pandemic context. As highlighted by [36], consumers now have access to real-time information and can choose where and how to shop, revolutionizing the multi-channel shopping experience. In this regard, companies need to develop effective strategies to manage and influence interactions with consumers across multiple touchpoints to drive their business forward.

4.2. Implications for Theory and Practice

The achieved results have implications for both theory and practice. From a theoretical perspective, the findings of this study have important implications for the field of marketing, especially in the post-pandemic context. By exploring the challenges faced by startups in developing their marketing plans, this study contributes to the advancement of literature and aligns with previous research [22]. The study's focus on innovation and validation of these challenges provides valuable insights into the specific demands that these companies face, greatly contributing to their recovery. The validated challenges also align with existing literature by emphasizing the importance of overcoming these aspects for the success of startups in the current landscape. Furthermore, the study offers important approaches to address the necessary changes in marketing strategies to navigate ongoing socioeconomic and technological transformations. These findings enrich the theoretical understanding of marketing practices in the current context and establish a solid foundation for overcoming the challenges faced in constructing startup marketing plans.

From a practical standpoint, the conducted study helps organizations gain a better understanding of how to deal with challenges and uncertainties during scenarios like the COVID-19 pandemic. This information assists companies in ensuring their survival and effectively harnessing the power of innovation and differentiation, thus achieving competitiveness and market presence. Furthermore, it provides a foundation for marketing man-agers to develop plans that align with the new challenges present in the market, taking into account the specificities of startups.

Reviewer 2 Report

Thank you for giving me an opportunity to review this interesting paper. I think this paper is well-written and informative for marketing practitioners as well as researchers. Here is my comment.

1. Title: Since this paper is based on an expert review in Brazil, the authors should add some words highlighted "Brazilian context" in the paper title.

2. The authors conducted Lawshe method for validation, but there is no concrete description of: what it is, how it is effective, and why it is needed.

3. The following sentences should be deleted.

"The Materials and Methods should be described with sufficient details to allow oth-ers to replicate and build on the published results. Please note that the publication of your manuscript implicates that you must make all materials, data, computer code, and proto-cols associated with the publication available to readers. Please disclose at the submission stage any restrictions on the availability of materials or information. New methods and protocols should be described in detail while well-established methods can be briefly de-scribed and appropriately cited."

4. The authors need to recheck the definition of transformative marketing.

Author Response

Rebuttal Letter

Manuscript ID: fintech-2533358

Dear reviewer,

Thank you very much for the considerations presented. These considerations have enabled a considerable improvement in our manuscript. We adjust according to your considerations. We list below all the considerations made and justify the actions taken. Thanks for your precious help.

Rebuttal items

Reviewer #2: Thank you for giving me an opportunity to review this interesting paper. I think this paper is well-written and informative for marketing practitioners as well as researchers. Here is my comment:

Reply: Thanks for the comment.

  1. Title: Since this paper is based on an expert review in Brazil, the authors should add some words highlighted "Brazilian context" in the paper title.

Reply: Thanks for the comment. The insertion was made. Please check the new version of the article.

  1. The authors conducted Lawshe method for validation, but there is no concrete description of: what it is, how it is effective, and why it is needed.

Reply: Thanks for the comment. Such explanations can be found in the methodological procedures section (see below).

“Afterward, the collected data was processed using the Lawshe method. This method was employed to analyze and validate the challenges identified in the research. This phase was carried out following the guidelines presented by [43] and [44] At first, the Content Validity Ratio (CVR) was calculated for each criterion in the questionnaire. The CVR values range from -1 to +1, where -1 indicates total disagreement and +1 indicates total agreement. When evaluating the unfavorable results obtained from the procedure, it is worth noting that a positive CVR was obtained when more than 50% of the respondents considered the analyzed items as "essential". Conversely, a negative CVR is obtained when less than 50% of the respondents deem the analytical item as "non-essential". When the CVR is equal to 0, it indicates that half of the experts consider the criterion as "necessary" while the other half does not [45]. Then calculations are performed. So, the calculation of 〖CVR〗_critical, is applied to analyze items that can be discarded in the final composition, due to CVR values below the critical limit. For the calculation in 〖CVR〗_critical, the parameters mean, variance and standard deviation are considered. All the aforementioned equations are presented in detail in Figure 1.”

  1. The following sentences should be deleted.

"The Materials and Methods should be described with sufficient details to allow others to replicate and build on the published results. Please note that the publication of your manuscript implicates that you must make all materials, data, computer code, and proto-cols associated with the publication available to readers. Please disclose at the submission stage any restrictions on the availability of materials or information. New methods and protocols should be described in detail while well-established methods can be briefly de-scribed and appropriately cited."

Reply: Thank for the comment. It was excluded. This part is a residue of text left over from the journal formatting template. We apologize for what happened. Was removed. Please check this new version of the paper.

  1. The authors need to recheck the definition of transformative marketing.

Reply: Thank for the comment. However, changing the definition of transformative marketing initially used in the text can completely invalidate the research results, as the responding professionals assessed their opinions according to the definitions presented.

Reviewer 3 Report

Congrats for your paper, please proceed to these minor revisions to improve the quality of the paper:

1.       Add more substantial keywords and drop keywords such as “challenges”; or include the full meaning “Challenges in the Marketing Plan Development”

2.       Revise the English language carefully: for example, at figure 1, there a few words in Portuguese

3.       To improve readability, you can write the six validated challenges at the beginning of each paragraph and bold them

4.       Develop a few more lines on the originality and main contributions of your paper: for example: are these challenges valid for all organizations? How can these challenges be contextualized?

 Revise the English language carefully: for example, at figure 1, there a few words in Portuguese

Author Response

Rebuttal Letter

Manuscript ID: fintech-2533358

Dear reviewer,

Thank you very much for the considerations presented. These considerations have enabled a considerable improvement in our manuscript. We adjust according to your considerations. We list below all the considerations made and justify the actions taken. Thanks for your precious help.

Rebuttal items

Reviewer #3: Congrats for your paper, please proceed to these minor revisions to improve the quality of the paper:

Reply: Thanks for the comment.

  1. Add more substantial keywords and drop keywords such as “challenges”; or include the full meaning “Challenges in the Marketing Plan Development”.

Reply: Thanks for the comment. Changes and insertions made. Please check the new version of the article.

  1. Revise the English language carefully: for example, at figure 1, there a few words in Portuguese.

Reply: Thanks for the comment. In this new version of the paper, a general revision of the English was made, as well as of Figure 1. Please check the new version of the article.

  1. To improve readability, you can write the six validated challenges at the beginning of each paragraph and bold them.

Reply: Thank for the comment Due to the journal's training rules, we cannot make this change. However, we clarify that each of the 6 validated challenges is discussed in section 4.1.

  1. Develop a few more lines on the originality and main contributions of your paper: for example: are these challenges valid for all organizations? How can these challenges be contextualized?

Reply: Thank for the comment. This information is at the conclusion of the article. See below.

“Therefore, it can be concluded that these challenges reflect the main concerns and obstacles faced by startups in building effective marketing strategies and striving for a competitive position in the market. By recognizing and understanding these challenges, startups will be better pre-pared to face adversities and seize opportunities in this new market context.”

It is important to note that this study has some limitations. The main limitation is related to the exploratory nature of the research, which implies a specific and restricted sample of marketing specialists operating in Brazil. Therefore, although a diverse sample of professionals from the field was obtained, including specialists from different regions with varied positions and experiences, it is important to consider that the results cannot be generalized to other contexts with different specificities.

However, despite the mentioned limitations, this research provides a solid foundation for future investigations in this area. Based on the results obtained and the identified gaps, several proposals for future research can be suggested. Firstly, it would be interesting to conduct a longitudinal study to track the evolution of the identified challenges over time, allowing for a deeper understanding of changes and trends in the post-pandemic context. Additionally, it would be valuable to investigate the specific strategies and approaches adopted by successful startups to overcome these challenges, aiming to identify best practices and insights that can be shared with other companies in similar situations. Another relevant point would be to explore the role of partnerships and collaborations between startups, established companies, and support institutions in addressing challenges and fostering innovation in the marketing sector. Lastly, considering the growing importance of corporate social responsibility, future research could explore the influence of sustainability and social responsibility practices on the marketing strategies of startups, particularly in the post-pandemic context. Such future research can contribute to the improvement of marketing strategies for startups and the development of an environment more conducive to their growth and success.”
